# Bis(maltolato)oxovanadium(IV) Induces Angiogenesis via Phosphorylation of VEGFR2

**DOI:** 10.3390/ijms21134643

**Published:** 2020-06-30

**Authors:** Laura Parma, Hendrika A.B. Peters, Maria E. Johansson, Saray Gutiérrez, Henk Meijerink, Sjef de Kimpe, Margreet R. de Vries, Paul H.A. Quax

**Affiliations:** 1Department of Surgery, Leiden University Medical Center, 2300 RC Leiden, The Netherlands; l.parma@lumc.nl (L.P.); H.A.B.Peters@lumc.nl (H.A.B.P.); m.r.de_vries@lumc.nl (M.R.d.V.); 2Einthoven Laboratory for Experimental Vascular Medicine, Leiden University Medical Center, 2300 RC Leiden, The Netherlands; 3Department of Physiology, Institute of Neuroscience and Physiology, University of Gothenburg, 43141 Gothenburg, Sweden; maria.e.johansson@neuro.gu.se (M.E.J.); saray_guti@hotmail.com (S.J.); 4CFM Pharma BV, 3401MA IJsselstein, The Netherlands; h.meijerink@cfmpharma.com (H.M.); s.dekimpe@cfmpharma.com (S.d.K.)

**Keywords:** angiogenesis, PTP inhibitor, VEGFR2

## Abstract

VEGFR2 and VEGF-A play a pivotal role in the process of angiogenesis. VEGFR2 activation is regulated by protein tyrosine phosphatases (PTPs), enzymes that dephosphorylate the receptor and reduce angiogenesis. We aim to study the effect of PTPs blockade using bis(maltolato)oxovanadium(IV) (BMOV) on in vivo wound healing and in vitro angiogenesis. BMOV significantly improves in vivo wound closure by 45% in C57BL/6JRj mice. We found that upon VEGFR2 phosphorylation induced by endogenously produced VEGF-A, the addition of BMOV results in increased cell migration (45%), proliferation (40%) and tube formation (27%) in HUVECs compared to control. In a mouse ex vivo, aortic ring assay BMOV increased the number of sprouts by 3 folds when compared to control. However, BMOV coadministered with exogenous VEGF-A increased ECs migration, proliferation and tube formation by only 41%, 18% and 12% respectively and aortic ring sprouting by only 1-fold. We also found that BMOV enhances VEGFR2 Y951 and p38MAPK phosphorylation, but not ERK1/2. The level of phosphorylation of these residues was the same in the groups treated with BMOV supplemented with exogenous VEGF-A and exogenous VEGF-A only. Our study demonstrates that BMOV is able to enhance wound closure in vivo. Moreover, in the presence of endogenous VEGF-A, BMOV is able to stimulate in vitro angiogenesis by increasing the phosphorylation of VEGFR2 and its downstream proangiogenic enzymes. Importantly, BMOV had a stronger proangiogenic effect compared to its effect in coadministration with exogenous VEGF-A.

## 1. Introduction

Angiogenesis is a key process in which new blood vessels grow into ischemic areas and it is mainly defined as the sprouting of new capillaries from existing blood vessels. The process of angiogenesis is regulated by proangiogenic growth factors that act via a family of endothelial receptor tyrosine kinases (RTKs) [1]. Binding of a growth factor to its receptor results in activation of the intracellular kinase domain and autophosphorylation of the receptor on specific tyrosine residues [2]. These events are followed by phosphorylation and activation of different downstream key angiogenic enzymes.

At a cellular level the angiogenic process involves three distinct cell functions: cell-migration, cell proliferation and finally formation of a new mature vessel [3]. These cell functions are carried out by phenotypically different endothelial cells (ECs), respectively: tip cells, stalk cells and phalanx cells [3]. This multistage process involves different RTKs [3,4].

The vascular endothelial growth factor (VEGF) family is composed of five structurally related factors: VEGF-A (also denoted VEGF-A165), VEGF-B, VEGF-C, VEGF-D and placenta growth factor (PLGF). VEGFs act through three structurally related VEGF receptor tyrosine kinases, denoted VEGFR1 (Flt1), VEGFR2 (Flk1) and VEGFR3 (Flt4) [5]. VEGF-A induces angiogenesis via stimulating endothelial cell migration and proliferation, mainly through the binding to the RTK vascular endothelial growth factor receptor 2 (VEGFR2) [6]. Binding of VEGF-A to VEGFR2 induces receptor dimerization and autophosphorylation at multiple tyrosine sites, including Y951, as well as the activation of downstream proangiogenic enzymes such as extracellular signal-regulated kinase (ERK)1/2 and p-38 mitogen-activated protein kinase (MAPK) regulated by the VEGFR2 phosphorylation sites Y1175 and Y1214 respectively [7]. Each tyrosine autophosphorylation site is thought to promote unique downstream signaling pathways, which are linked to different cellular responses such as proliferation, migration and permeability [8].

Tyrosine phosphorylation is controlled by an equilibrium between activation of protein tyrosine kinases and protein tyrosine phosphatases (PTPs). PTPs are a family of endogenous modulators of RTKs-mediated signaling pathways that carry out the dephosphorylation of phospho-tyrosine residues. PTPs act by either direct dephosphorylation of particular receptor tyrosine residues or of downstream signaling components [9]. Therefore, the blockade of PTPs would be a strategy to increase RTKs activation and subsequently angiogenesis.

The process of angiogenesis, together with arteriogenesis, represents a possible therapeutic strategy to treat patients affected by peripheral arterial disease (PAD) by restoring blood flow to ischemic tissues.

Vanadium compounds are a group of molecules that act as nonselective tyrosine phosphatase inhibitors [10]. Their mechanism of action, the inhibition of PTPs, was first shown on the insulin receptor where vanadium compounds act as insulin-mimic enhancing its phosphorylation [11,12]. In this context bis(maltolato)oxovanadium(IV) (BMOV) was shown to have antidiabetic properties and insulin-mimicking effects and to improve cardiac dysfunctions in diabetic models [13,14,15]. Previously used as simple inorganic vanadium salts, they were then replaced by larger and more complex compounds with organic ligands, that have shown increased bioavailability [10,16,17].

In the present study we focus on the role of bis(maltolato)oxidovanadium (IV) (BMOV) (Figure 1B), an organic vanadium salt, in wound closure and in angiogenesis and its effect on VEGFR2 signaling, chosen as a representative of the RTKs due to its pivotal role in angiogenesis.

It has previously been shown that different organic vanadium salts have a positive effect in promoting wound healing closure, most likely via increasing angiogenesis. Vanadyl acetylacetonate (VAC) has been shown to increase chondrogenesis and angiogenesis in fracture healing in rats [18,19,20]. Based on these findings we hypothesized that BMOV would promote wound closure in vivo in C57BL/6JRj mice.

Moreover, based on our findings that human umbilical vein endothelial cells (HUVECs) endogenously produce a baseline amount of VEGF-A and this results in VEGFR2 phosphorylation, we hypothesized that the addition of BMOV would increase in vitro angiogenesis via blocking PTPs-induced receptor de-phosphorylation. Moreover we hypothesized that exogenous addition of VEGF-A would enhance the effect of BMOV resulting in increased VEGFR2 activation and subsequent increased angiogenesis (Figure 1A).

Therefore, in the present study we assessed the effect of BMOV on in vivo wound closure and the effect of BMOV and coadministration of BMOV and exogenous VEGF-A in different in vitro angiogenesis assays. Moreover, we examined which phospho-residues of VEGFR2 are involved in response to BMOV.

## 2. Results

### 2.1. BMOV Induces Wound Closure in vivo

To investigate whether BMOV affects wound healing in vivo we performed two independent wound healing experiments. As shown in Figure 2, 4 days after wound formation, mice treated with 5mg/kg BMOV showed a strong reduction in the wound area when compared to control mice treated with a saline solution. In fact the wound area of BMOV-treated mice was reduced by 45% when compared to control (Figure 2, *p* < 0.001).

### 2.2. HUVECs Produce Endogenous VEGF-A and this is not Affected by BMOV Treatment

HUVECs endogenously produce low levels of VEGF-A (51 pg/mL) and the amount of VEGF-A in HUVECs that were treated with BMOV for 12 h was similar to the amount in the nontreated control cells (Figure 3, *p* = 0.14). When looking at the amount of VEGF-A in the culture media of cells treated with exogenous VEGF-A (10 ng/mL) or VEGF-A coadministered with BMOV, no differences could be observed (Figure 3, *p* = 0.35).

### 2.3. Endothelial Cell Migration is Induced by BMOV Treatment

To understand how BMOV affects the first step toward the formation of a new vessel, we examined its effect on the migration of endothelial cells after 18 h of treatment. HUVECs treated with increasing doses of BMOV showed dose-dependent enhanced scratch-wound closure when compared to untreated control cells and this induction reached a significant difference with the highest dose tested (Figure 4A and Appendix A, *p* = 0.009).

Quantification of the migration rate showed an increase in cell migration by 45% in the group treated with 15 µM BMOV when compared to control (Figure 4A and Appendix A).

The effect of coadministration of BMOV and VEGF-A on ECs migration was assessed by adding to the cell culture media 10 ng/mL VEGF-A and/or increasing doses of BMOV, 0.5, 5 and 15 µM respectively. In this set-up coadministration of BMOV and VEGF-A was able to increase cell migration rate, reaching a 78% enhanced migration compared to control (10 ng/mL VEGF-A) using exogenous VEGF-A together with BMOV at the dose of 15 µM (Figure 4B and Appendix A).

However, the effect seen in Figure 4B resulted to be the sum of the effect of BMOV and the effect of VEGF-A. In fact, BMOV gave a 45% increase in cell migration compared to control (untreated cells) (Figure 4A) and when co-administered with VEGF-A the isolated effect of the highest BMOV dose tested resulted in a 41% increase in migration on top of the VEGF-A effect (Figure 4C, *p* = 0.01) compared to cells treated with 10 ng/mL VEGF-A.

### 2.4. BMOV Induces ECs Proliferation

To determine whether BMOV had an effect on ECs proliferation and what would happen to this effect when extra VEGF-A is added to the cells, an MTT assay was performed and cell proliferation was evaluated after 24 h of treatment.

All the doses of BMOV tested (0.5, 5 and 15 µM) resulted in a significant increase in ECs proliferation (*p* = 0.02, 0.01 and 0.008 respectively) reaching a 42% increase in proliferation with the dose of 15 µM when compared to control (Figure 5A and Appendix A).

To examine if BMOV and exogenously added VEGF-A work together toward increasing cell proliferation, we repeated the experiment as above coadministering 10 ng/mL VEGF-A and/or BMOV in different doses to the cell culture. As shown in Figure 5B and Appendix A, cells incubated with BMOV supplemented with exogenous VEGF-A showed an increased proliferation rate compared to cells treated with only 10 ng/mL VEGF-A. BMOV (5 µM ) with the addition of 10 ng/mL VEGF-A resulted in an increase in cell proliferation by 27% when compared to control represented by 10 ng/mL VEGF-A (*p* = 0.02).

As shown in Figure 5C the effect of BMOV in the experiment in which we coadministered it with exogenous VEGF-A resulted to be less pronounced when compared to the experiment in which we added only BMOV to cells. In fact, in Figure 5A, BMOV-treated cells reached an increase of 42% in proliferation when compared to control (untreated cells), while with the supplementation of exogenous VEGF-A cell proliferation induced by BMOV increased by only 18% when compared to control (cells treated with exogenous VEGF-A) (Figure 5C). Therefore, the effect seen in Figure 5B is the sum of individual effects of BMOV and VEGF-A.

### 2.5. Tube Formation is Stimulated by BMOV Treatment 

To evaluate whether BMOV can increase the capability of ECs to form capillary-like structures a tube-formation assay was performed and the total length of tubes formed was determined after 12 h of treatment (Figure 6A and B). BMOV was able to promote HUVECs tube formation by 12% at a dose of 5 µM and by 27% at 15 µM (Figure 6A and Appendix A, *p* = 0.003) compared to control (untreated cells).

The effect of coadministration of the highest dose of BMOV and exogenous VEGF-A on tube formation in HUVECs resulted in a 20% increase in total length of the tube formed (Figure 6B and Appendix A) when compared to control (cells treated with 10 ng/mL VEGF-A). When coadministering BMOV and VEGF-A, the isolated additive effect of BMOV was a 12% increase in tube formation (Figure 6C) on top of the effect of VEGF-A (Figure 6B).

### 2.6. Aortic Ring Sprouting is Induced upon BMOV Stimulation

To examine the effects of BMOV on the sprouting of capillaries we used an ex vivo mouse aortic ring assay, in which not only endothelial cells are present but also other cell types, including smooth muscle cells.

A 2- and 3-fold increase in the number of sprouting neovessels formed in the aortic segments treated respectively with 5 and 15 µM BMOV were found when compared to untreated control (Figure 7A; *p* = 0.004 and 0.0002, respectively). Cell-type-specific double staining of aortic segments confirmed that the sprouting neovessels are lined with endothelial cells, as described previously [21,22]. The endothelial cells in the sprouts were surrounded by smooth muscle cells, characteristic of mature neovessels (Figure 7A and Appendix A).

The same vessel structure was observed when co-administering exogenous VEGF-A and increasing doses of BMOV (0.5, 5 and 15 µM) to the cultured rings. Endothelial cells formed the neovessels sprouting from the rings, and they were surrounded by smooth muscle cells (Figure 7B and Appendix A).

The addition of VEGF-A resulted in almost a 4-fold increase in the number of sprouts when compared to control represented by untreated aortic segments (Figure 7B).

Coadministration of VEGF-A and BMOV in increasing doses enhanced the average number of sprouts formed when compared to the untreated control cultured rings (Figure 7B). However, when administered together with VEGF-A, the isolated effect of BMOV resulted in an increase in sprouting neovessels only with the 5 µM dose when compared to control (aortic segments treated with 10 ng/mL VEGF-A) (28% increase), while the other doses did not show any effect (Figure 7C).

### 2.7. BMOV acts via VEGFR2 

As unspecific blocker of protein phosphatases BMOV has been shown to exploit its action on several receptors’ downstream signaling [11]. Due to its pivotal role in angiogenesis we focused on the effect of BMOV on VEFGR2 downstream signaling. To do so we analyzed the phosphorylation and therefore the activation of one of the main VEGFR2 phosphorylation sites (Y951) and two key angiogenic enzymes, ERK1/2 and p-38MAPK regulated by the phosphorylation sites Y1175 and Y1214 respectively (Figure 8A).

The amount of VEGFR2 phosphorylated on the residue Y951 was increased by 1.5 fold in the group treated with 15 µM BMOV when compared to control (untreated cells) (Figure 8B) (*p* = 0.04). The phosphorylation on the same residue in the groups costimulated with 15 µM BMOV and exogenous VEGF-A showed no differences between the groups (*p* = 0.36).

The phosphorylation of ERK1/2 (Figure 8C), downstream signaling of Y1175, showed no differences between control and treated groups. The groups treated with coadministration of exogenous VEGF-A and BMOV showed an increase in phosphorylation of ERK1/2 when compared to control (untreated cells) but no differences were found between BMOV-treated and respective controls (untreated cells and cells treated with 10 ng/mL VEGF-A. *p* = 0.96 and *p* = 0.99 respectively) (Figure 8C).

An almost significant increase (*p* = 0.05) was found in the activation of p38, downstream of Y1214 signaling. BMOV treatment resulted in a 1.2-fold increase in p38 phosphorylation when compared to the control (untreated cells) (*p* = 0.05. Figure 8D). Coadministration of VEGF-A and BMOV did not show differences when compared to each other (*p* = 0.8), even though they were 1.4-fold higher than the control (untreated cells) (Figure 8D).

## 3. Discussion

In the present study BMOV increases in vivo wound closure and in vitro endothelial cell migration, proliferation and tube formation in HUVECs. Additionally, it increases the number of sprouts formed in an ex vivo aortic ring assay. BMOV increases phosphorylation of VEGFR2 via Y951 and p38MAPK, but not ERK1/2 phosphorylation. We also show that BMOV and exogenous VEGF-A do not work in a synergistic way to increase angiogenesis.

Research from Paglia et al. and Ippolito et al. has shown that a different organic vanadium salt, Vanadyl acetylacetonate (VAC), has a positive effect in promoting chondrogenic wound healing closure, most likely via increasing angiogenesis [18,19,20]. It was previously described that BMOV could enhance collateral blood flow in a rat model of peripheral vascular disease and increase the diameter of cerebral collateral in rats [9,23]. Interestingly, these latter studies focus primarily on collateral maturation and therefore arteriogenesis, whereas patients with peripheral arterial disease (PAD) benefit by increased blood flow recovery via a combined action of improved angiogenesis and arteriogenesis [24]. Therefore, in this study we examined the role of BMOV in angiogenesis. We show that BMOV stimulates in vivo wound closure and increases in vitro angiogenesis via acting on the whole angiogenic process including the migration and proliferation of ECs and their ability to form a new vessel. Combined, these results indicate that BMOV has an action on both processes of angiogenesis and arteriogenesis.

In the context of PAD, although very promising, VEGF therapy did not show the expected clinical results in patients affected by PAD [25,26]. VEGF was shown to be present and even increased at the site of amputation in patients with critical ischemic PAD but it was not effective enough to restore blood flow and induce angiogenesis [25,26]. These findings show that there is a need for new therapeutic options for the treatment of PAD. Here, we demonstrate that low doses of endogenous VEGF-A activate VEGFR2 and BMOV boosts this induced proangiogenic signaling cascade via PTPs blockade (Figure 9).

BMOV increases in vitro and ex vivo angiogenesis by increasing the number of neovessels in a complex angiogenic assay in which all the ECs functions come together toward the formation of a new vessel. Based on these preliminary results BMOV could be a new, interesting therapeutic option to induce angiogenesis in PAD patients in which a low dose of VEGF-A is present but it is not enough to completely restore angiogenesis.

Carr et al have previously shown that coadministered BMOV and exogenous VEGF-A increased HUVECs survival compared to exogenous VEGF-A administered alone [23]. We therefore examined if coadministration of BMOV and exogenous VEGF-A could result in increased angiogenesis. In line with the results from Carr et al we found that coadministration of BMOV and VEGF-A resulted in higher stimulation of in vitro angiogenesis compared to administration of exogenous VEGF-A. However we also found that this was not a synergistic but rather a cumulative effect resulting from the sum of individual BMOV and VEGF-A effects. More importantly, we found that the effect of BMOV when coadministered with exogenous VEGF-A resulted to be less strong in inducing angiogenesis compared to the effect of the single administration of BMOV. Except for the migration assay, in which the effect of BMOV coadministered with VEGF-A resulted to be similar to the effect of BMOV administered alone, in the cell proliferation assay, tube assay and aortic ring assay, the effect of BMOV resulted to be higher than its effect when used in combination with exogenous VEGF-A (Figure 8). The phosphorylation levels of VEGFR2 (Y951), ERK1/2 and p38MAPK, in HUVECs treated with exogenous VEGF-A and exogenous VEGF-A coadministered with BMOV resulted to be the same. Therefore, due to the fact that high VEGFR2 activation is achieved with exogenous addition of VEGF-A, it is likely that in this situation BMOV does not have a significant effect on angiogenesis induced by VEGFR2 activation because the receptor is already fully active.

In the present manuscript we show not only that VEGFR2 activation is augmented upon BMOV treatment but we also pinpoint the phospho-residues involved in the signaling. In the present set-up BMOV increased the activation of Y-951 and p38MAPK while ERK1/2 was not affected. It was previously shown that BMOV could enhance the phosphorylation of ERK1/2 in skeletal muscle extracts of diabetic rats, suggesting that our results are cell-type specific and dependent on our experimental setup [27]. More importantly Y-951, p38MAPK and ERK1/2 phosphorylation results are not completely in line with the results we obtained in the in vitro assays. It was previously shown that cell migration and sprout formation are regulated by Y-951 and p38MAPK activation, while cell proliferation depends on ERK1/2 activation [28,29]. We here found that BMOV increased HUVECs migration and sprout formation supported by phosphorylation of VEGFR2 Y-951 and p38MAPK, but BMOV also increased HUVECs proliferation and this was not supported by ERK 1/2 activation. As a potential limitation of our study, we used the MTT assay as a proliferation assay whereas it is an assay that measures the mitochondrial activity of the cells [30]. This could be the explanation why the results of our proliferation assay and the phosphorylation of ERK1/2 do not correlate.

Based on the results obtained in this study we can conclude that BMOV alone induces in vitro angiogenesis and does not act synergistically with VEGF in this process. Moreover, BMOV is able to activate VEGFR2 and downstream proangiogenic enzymes without exogenous addition of VEGF. Therefore, our results show that BMOV-mediated inhibition of PTPs is a promising strategy to induce angiogenesis.

## 4. Materials and Methods

### 4.1. In vivo Wound Healing

All procedures involving mice were approved by the Regional Animal Ethics Committee at the University of Gothenburg, in accordance with the European Communities Council Directives of 22 September 2010 (2010/63/EU).

Male C57BL/6JRj mice (Janvier Labs, Le Genest-Saint-Isle, France) were anesthetized with isoflurane, hair was removed by shaving using hair removal cream. All animals received Temgesic (0.08 mg/kg BW) prior to creating the wounds. Skin was wiped with 70% ethanol and thereafter two 5-mm-diameter wounds were created at the back of the mouse using a biopsy puncher. BMOV (5mg/kg) or saline was administered via intravenous (iv) injections in the tail vein. Wounds were initially covered with tegaderm band aid (3M, Apoteket AB, Stockholm, Sweden) for the first two days, to avoid sawdust and bedding material getting into the wounds, which were thereafter removed. Wounds were measured using a digital caliper at day 0, 2 and 4 and healing rate is expressed as percentage of initial area. Mice were sacrificed at day 4 by an overdose of pentobarbital (i.p., Apoteket AB).

Animals were housed at 21–24 °C in a room with 12-h light/12-h dark cycle. Water and food were available ad libitum.

### 4.2. Isolation of Human Umbilical Venous Endothelial Cells (HUVECs)

For the isolation of HUVECS anonymous umbilical cords were obtained in accordance with guidelines set out by the ‘Code for Proper Secondary Use of Human Tissue’ of the Dutch Federation of Biomedical Scientific Societies (Federa, Rotterdam, The Netherland), and conforming to the principles outlined in the Declaration of Helsinki. Isolation and culturing of primary venous human umbilical cells was performed as described by Van der Kwast et al. [21]. In brief, umbilical cords were collected from full-term pregnancies and stored in sterile PBS at 4 °C and subsequently used for cell isolation within 5 days. For HUVEC isolation, cannulas were inserted on each side of the vein of an umbilical cord and flushed with sterile PBS. The artery was infused with 0.075% collagenase type II (Worthington, Lakewood, NJ, USA) and incubated at 37 °C for 20 min. The collagenase solution was collected and the artery was flushed with PBS in order to collect all detached endothelial cells. The cell suspension was centrifuged at 400 g for 5 min and the pellet was resuspended in HUVEC complete culture medium (EBM-2 Basal Medium (CC-3156) and EGMTM-2 SingleQuotsTM Supplements (CC-4176), Lonza). HUVECs were cultured in plates coated with 1% fibronectin and used between passage 2 and 4. Low-serum culture medium consisted of EBM-2 Basal Medium (CC-3156, Lonza, Basel, Zwitserland) supplemented with 0.2% Fetal Bovine Serum (FBS) and 1% GA-1000 (SingleQuotsTM Supplements, CC-4176, Lonza).

### 4.3. BMOV Preparation

BMOV was kindly gifted by CFM pharma (Figure 2B).

For scratch-wound healing assay, cell proliferation assay, tube formation assay and aortic ring assay, bis(maltolato)oxovanadium(IV) (BMOV) was dissolved in PBS and used for the assays at a final concentration of 0.5, 5 and 15 µm.

For samples used for western blot and ELISA, BMOV was dissolved in PBS and used for the assay at a final concentration of 15 µm.

The selection of the doses of BMOV used for both in vitro and in vivo experiments was based on previous publications that used BMOV or vanadium derivatives in the context of arteriogenesis and wound healing [18,19,23].

### 4.4. Scratch-Wound Healing Assay

For the scratch-wound healing assay (*n* = 3 experimental replicates), HUVECs cells were plated on a 12-well plate and grown until 80% confluence in complete culture medium as previously reported [21]. Cells were then treated with low-serum medium or low serum supplemented with BMOV in increasing concentrations for 24 h. After 24 h, medium was removed and a scratch wound was introduced across the diameter of each well of a 12-well plate using a p200 pipette tip. Subsequently, the cells were washed with PBS and medium was replaced by new low-serum culture medium with or without 10 ng/mL VEGF-A (Human VEGF-A165, 718302, Biolegend, Amsterdam

Netherlands) and/or BMOV in increasing concentrations. Two locations along the scratch wound were marked per well and scratch-wound closure at these sites was imaged by taking pictures at time 0 h and 18 h after scratch-wound introduction using live phase–contrast microscopy (Axiovert 40C, Carl Zeiss, Oberkochen, Germany). Average scratch-wound closure after 18 h was objectively calculated per well by measuring difference in cell coverage at 18 h vs. 0 h using the wound healing tool macro for ImageJ.

### 4.5. MTT Assay

Cell proliferation (*n* = 3 experimental replicates) was measured using MTT assay. HUVECs cells were plated at 5000 cells/well in a 96-well plate and grown until 80% confluency in complete culture medium, after which they were incubated with low-serum medium or low serum supplemented with BMOV in increasing concentrations for 24 h. The medium was then replaced by treatment mixtures consisting of low-serum medium with or without 10 ng/mL VEGF-A, and/or BMOV in increasing concentrations. After 24 h incubation, 10 µL MTT (Thiazolyl blue tetrazolium bromide, Sigma M5655) was added directly to each well and cells were incubated at 37 °C in a humidified 5% CO2 environment for 4 h. Subsequently, 75 µL medium was removed from each well and 75 µL isopropanol/0.1N HCL was added per well. After incubating the plate for 90 min on a shaker platform, absorbance was read at 570 nm with a Cytation 5 spectrophotometer (BioTek) and the mitochondrial metabolic activity was quantified as a representative measure of cell proliferation. 

### 4.6. Tube Formation Assay

HUVECs were seeded in 12-well plates in complete medium until confluent. Medium was then replaced with low-serum culture medium or low serum supplemented with BMOV in increasing concentrations for 24 h. A 96-wells plate was coated using 50 µL/well of Geltrex extracellular matrix (A1413202, Gibco, Waltham, USA). Cells were then detached using trypsin-EDTA (Sigma, Steinheim, Germany) and diluted at a concentration of 150.000 cells/mL in low-serum medium with or without 10 ng/mL VEGF-A and/or BMOV in increasing concentrations. After 12 h incubation pictures of each well were taken using live phase–contrast microscopy (Axiovert 40C, Carl Zeiss, Oberkohen, Germany). Total length of the tubes formed was analyzed using the wound healing tool macro for ImageJ (NIH and LOCI, University of Wisconsin, WI, USA)

### 4.7. Quantification of VEGF-A in Cell Culture Medium

HUVECs were seeded in 12-well plates in 1 mL of complete medium for 24 h until confluent. Medium was then replaced with low-serum culture medium for an additional 24 h. Cells were stimulated in low-serum medium with or without 10 ng/mL VEGF-A, and/or 15µm BMOV for 60 min. Cell culture medium was collected and stored at -80°C. VEGF-A concentration was measured by a sandwich ELISA (DY293B-05, R&D Systems, Minneapolis, MN, USA) according to the manufacturer’s instructions.

### 4.8. Aortic Ring Assay

Mouse aortic ring assay was performed as described previously [31]. In brief, six thoracic aortas were removed from 8- to 10-week old mice, after which the surrounding fat and branching vessels were carefully removed and the aorta was flushed with Opti-MEM medium (Gibco, Thermo Fisher Scientific, Waltham, MA, USA).

Aortic rings of 1 mm were cut and the rings from each mouse aorta were incubated overnight with fresh Opti-MEM.

96-well plates were coated with 75 μL collagen matrix (Collagen (Type I, Millipore, Burlington, MA, USA) diluted in 1x DMEM (Gibco, Thermo Fisher Scientific, Waltham, MA, USA ) and pH-adjusted with 5N NaOH. One aortic ring per well was embedded in the collagen matrix, for a total of 30 rings per condition. After letting the collagen solidify for one hour, 150 μL Opti-MEM supplemented with 2.5% FBS (PAA, Pasching, Austria), 1% penicillin-streptomycin (PAA, Pasching, Austria) and with or without 10 ng/mL VEGF-A (R&D systems, Minneapolis, MN, USA) and/or either 0.5, 5 or 15µm BMOV was added to each well. Medium was refreshed every other day. Pictures of each embedded aortic ring and their neovessel outgrowth were taken after 7 days using live phase–contrast microscopy (Axiovert 40C, Carl Zeiss, Oberkochen, Germany). The number of neovessel sprouts were counted manually. Each neovessel emerging from the ring was counted as a sprout and individual branches arising from each microvessel counted as a separate sprout. For immunohistochemistry the embedded rings were formalin fixed and permeabilized with 0.25% Triton X-100. A triple staining was performed using primary antibodies against smooth muscle cells (α-smooth muscle actin, 1A4, DAKO), endothelial cells (CD31, BD Pharmingen, San Diego, USA) and nuclei (DAPI). Alexa Fluor 647, Alexa Fluor 488 antibodies (Life Technologies) were used as secondary antibodies and slides were mounted with ProLong Gold mountant with DAPI (P36935, ThermoFisher, Waltham, USA). Stained slides were photographed using a Panoramic MIDI II digital slide scanner (3DHISTECH).

### 4.9. Sample Preparation and Western Blot

HUVECs were seeded in 12-well plates in complete medium for 24 h until confluent. Medium was then replaced with low-serum culture medium for additional 24 h. Cells were stimulated in low-serum medium with or without 10 ng/mL VEGF-A, and/or 15 µm BMOV in PBS for 60 min. Cells were then lysated using modified RIPA buffer [10 mM Tris-HCl pH = 7.4 (10708976001, Sigma-Aldrich, Saint Louis, USA), 150 mM NaCl (S7653, Sigma-Aldrich), 5 mM EDTA pH = 8.0 (E9884, Sigma-Aldrich), 1% Triton X-100, 1% SDS (L3771, Sigma-Aldrich), 1 mM NaF (S7920, Sigma-Aldrich), 1mM Na3VO4 (S6508, Sigma-Aldrich) and cOmplete™ Protease Inhibitor Cocktail (1169749800, Roche Diagnostics)].

Western blot was performed as described by Van der Kwast et al. [32]. Total protein concentration was quantified using a Pierce BCA ProteinAssay Kit (Thermo Fisher Scientific), after which protein concentration was normalized to 1.25 mg/mL in Laemmli buffer (Bio-Rad Laboratories, Hercules, CA, USA) containing 10%b-mercaptoethanol (Sigma-Aldrich). Proteins were separated in the Vertical Electrophoresis Cell system using pre-mixed Tris/glycine/SDS running buffer (Bio-Rad Laboratories) and were transferred onto a nitrocellulose membrane (GE Health-care Life Sciences, Eindhoven, the Netherlands) using premixed Tris/glycine transfer buffer (Bio-Rad Laboratories, Hercules, USA). Blots were incubated overnight at 4°C with antibodies directed either at p-VEGFR2 (2476s, cell signaling), or p-ERK1/2 (M8159, Sigma-Aldrich) or p-p38 (92115, cell signaling) or stable household protein vinculin (V9131, Sigma-Aldrich) or β-actin (Ab8224, AbCam, Cambridge, UK) diluted to 1:1000 in 5% BSA in TBS-T. The membrane was then incubated with either antimouse (31432, Thermo Fisher Scientific) or antirabbit antibody (31462, Thermo Fisher Scientific) peroxidase-conjugated secondary antibody diluted to 1:10.000 in 5% BSA in TBS-T. Proteins of interest were revealed using SuperSignal West Pico PLUS Chemiluminescent Substrate (ThermoFisher Scientific) and imaged using the ChemiDoc TouchImaging System (Bio-Rad Laboratories). p-VEGFR2 expression was quantified relative to stable household protein β-actin and p-ERK1/2 and p-p38 expression were quantified relative to stable household protein vinculin using ImageJ.

### 4.10. Statistical Analysis

For the in vivo wound healing experiment, to adjust for sacrifice day, data from the two independent experiments, were analyzed and expressed as estimated marginal means ± SEM (IBM SPSS Statistics for windows, Version 25.0, Armonk, NY:IBM Corp).

Results of in vitro assays are expressed as mean ± SEM. A One-Way ANOVA or unpaired T-test were used to compare individual groups. Non-Gaussian distributed data were analyzed using a Kruskal-Wallis test using GraphPad Prism version 6.00 for Windows (GraphPad Software). Probability-values < 0.05 were regarded significant.

## Figures and Tables

**Figure 1 ijms-21-04643-f001:**
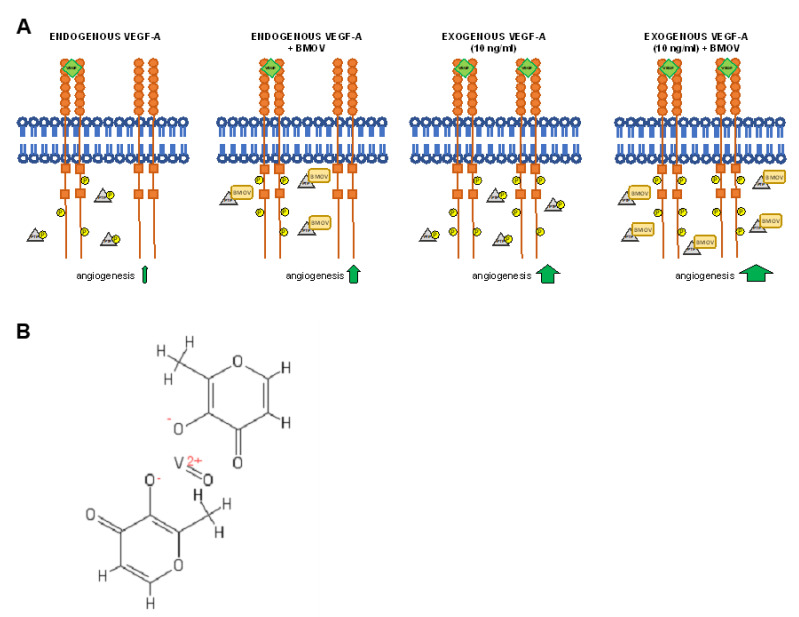
(**A**) Schematic representation of VEGFR2 signaling activation in response to the indicated conditions. (**B**) bis(maltolato)oxovanadium(IV) (BMOV) chemical structure.

**Figure 2 ijms-21-04643-f002:**
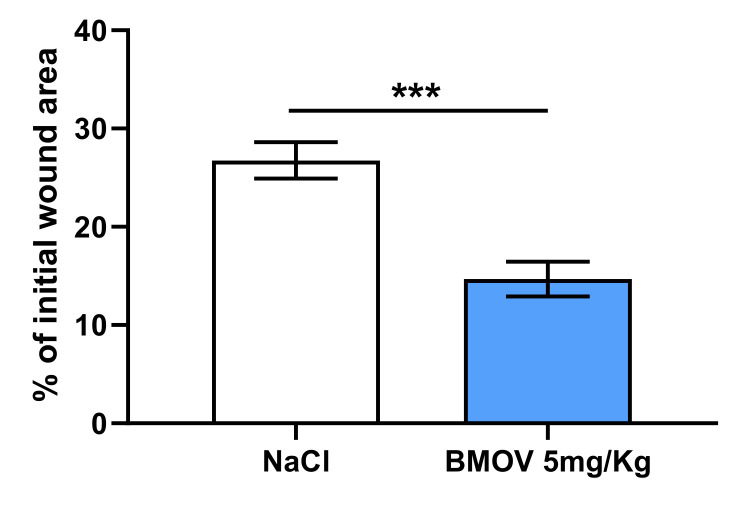
Quantification of in vivo wound area after 4 days’ treatment with either saline solution or 5mg/kg BMOV. Wound healing data from two independent experiments were pooled and expressed as estimated marginal means ± SEM. ***p < 0.001, *n* = 11-12/group.

**Figure 3 ijms-21-04643-f003:**
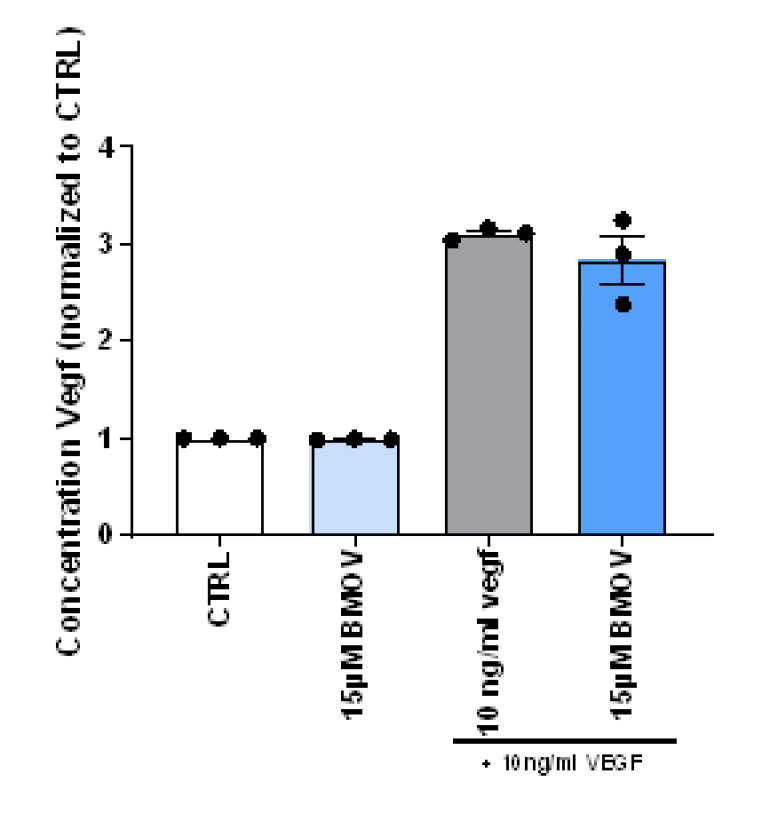
Quantification of the concentration of VEGF-A concentration in the cell culture medium of HUVECs incubated with the indicated conditions. All data points represent normalized averages obtained from 3 independent experiments and are presented as mean ± SEM. Two-sided Student’s t test to compare control versus BMOV treatments.

**Figure 4 ijms-21-04643-f004:**
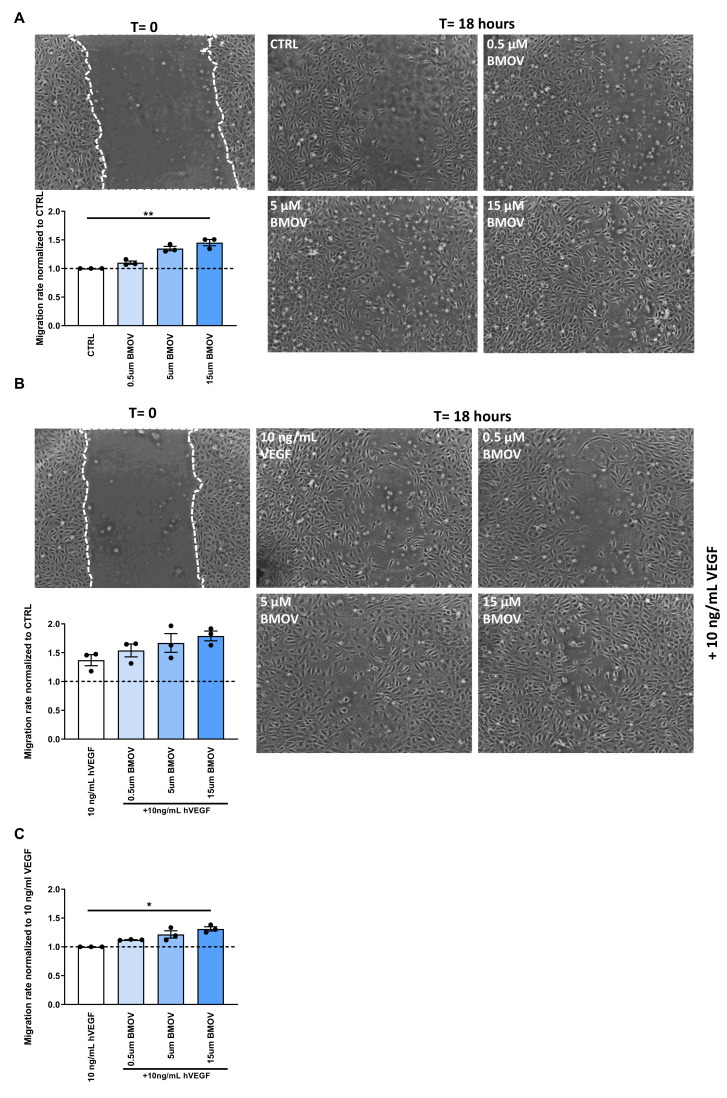
Representative images and quantification of HUVECs scratch-wound healing after treatment with either (**A**) BMOV alone in different concentrations (0.5, 5, 15 µM) or (**B**) different concentrations of BMOV (0.5, 5, 15 µM) supplemented with 10 ng/mL VEGF-A. (**C**) Effect of BMOV in the cell culture supplemented with 10 ng/mL VEGF-A. Datapoints represent averages obtained from 3 independent experiments and are presented as mean ± SEM. **p* < 0.05; ***p* < 0.01 by Kruskal-Wallis test.

**Figure 5 ijms-21-04643-f005:**
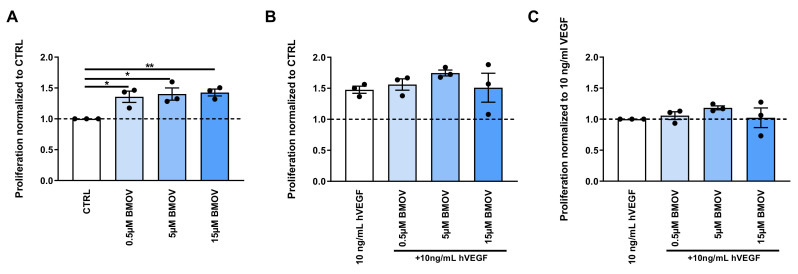
Quantification of HUVECs proliferation after treatment with either (**A**) BMOV alone in different concentrations (0.5, 5, 15 µM) or (**B**) different concentrations of BMOV (0.5, 5, 15 µM) supplemented with 10 ng/mL VEGF-A. (**C**) Effect of BMOV in the cell culture supplemented with 10 ng/mL VEGF-A. Datapoints represent averages obtained from 3 independent experiments and are presented as mean ± SEM. **p* < 0.05; ***p* < 0.01 by One-Way ANOVA.

**Figure 6 ijms-21-04643-f006:**
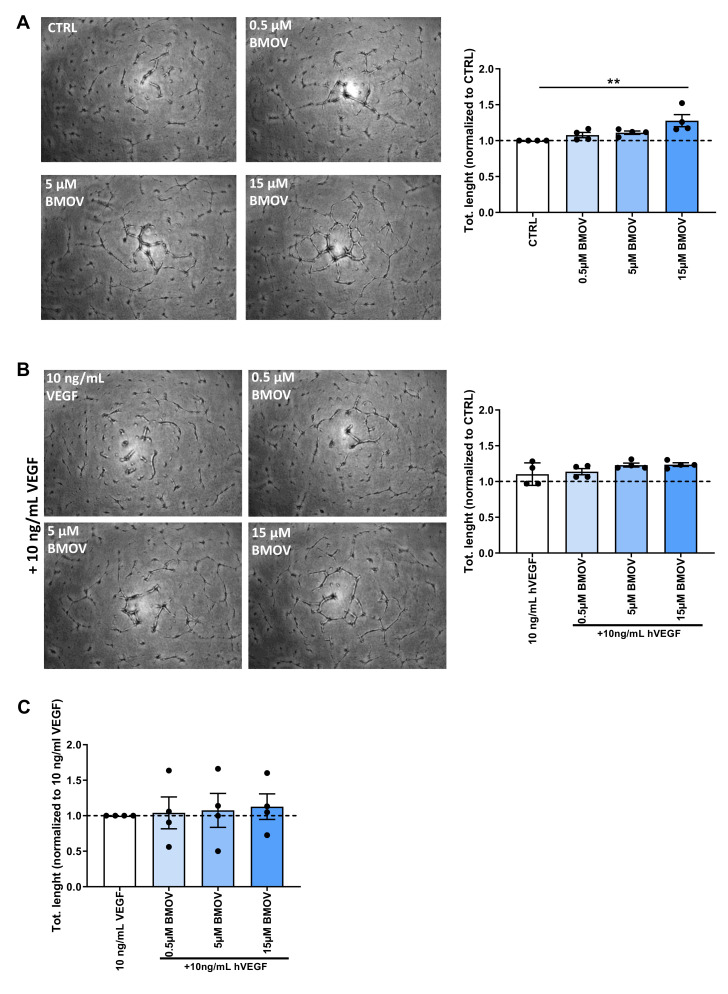
Representative images and quantification of HUVECs tube formation after treatment with either (**A**) BMOV alone in different concentrations (0.5, 5, 15 µM) or (**B**) different concentrations of BMOV (0.5, 5, 15 µM) supplemented with 10 ng/mL VEGF-A. (**C**) Effect of BMOV in the cell culture supplemented with 10 ng/mL VEGF-A. Datapoints represent averages obtained from 3 independent experiments and are presented as mean ± SEM. ***p* < 0.01 by One-Way ANOVA.

**Figure 7 ijms-21-04643-f007:**
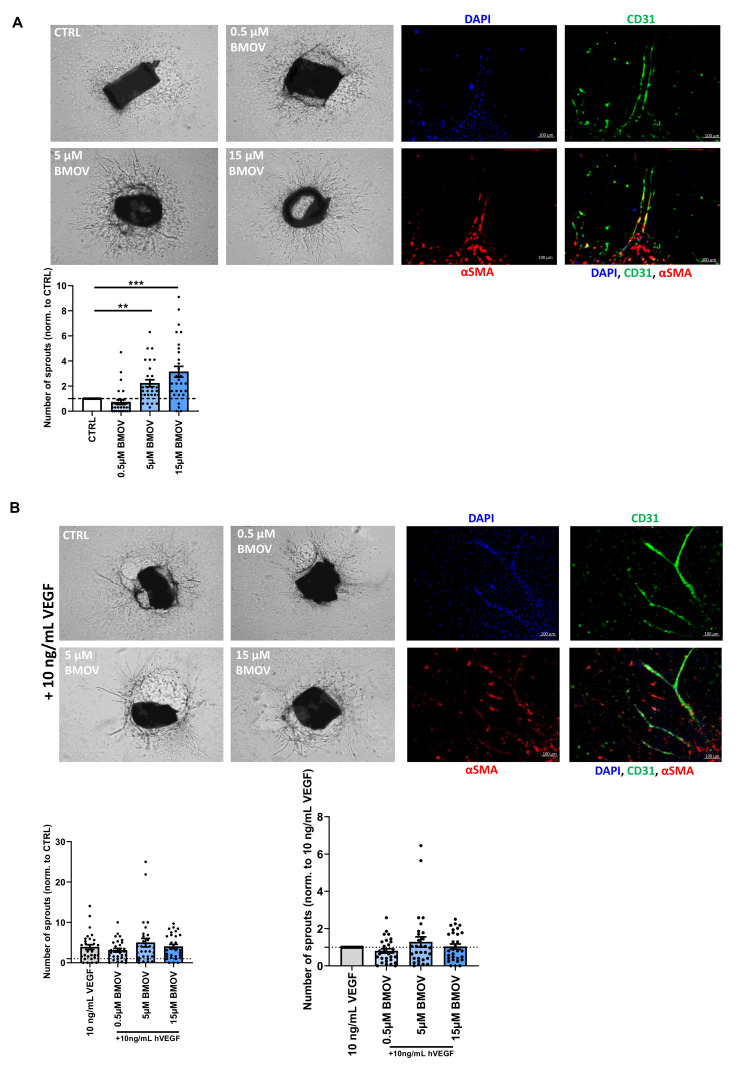
Representative images (top) and quantification of neovessel sprouts (bottom) after treatment with either (**A**) BMOV alone in different concentrations (0.5, 5, 15 µM) or (**B**) different concentrations of BMOV (0.5, 5, 15 µM) supplemented with 10 ng/mL VEGF-A in an ex vivo aortic ring assay. Top right panels of (**A**,**B**) are representative examples of complex aortic ring neovessel sprouts fluorescently stained with cell specific markers. CD31 (green) stains endothelial cells; α-SMA (red) indicates smooth muscle cells; DAPI-stained nuclei (blue). (C) Effect of BMOV in the tissue culture supplemented with 10 ng/mL VEGF-A. Datapoints are presented as mean ± SEM. ***p* < 0.01; *** *p* < 0.001 by Kruskal-Wallis test.

**Figure 8 ijms-21-04643-f008:**
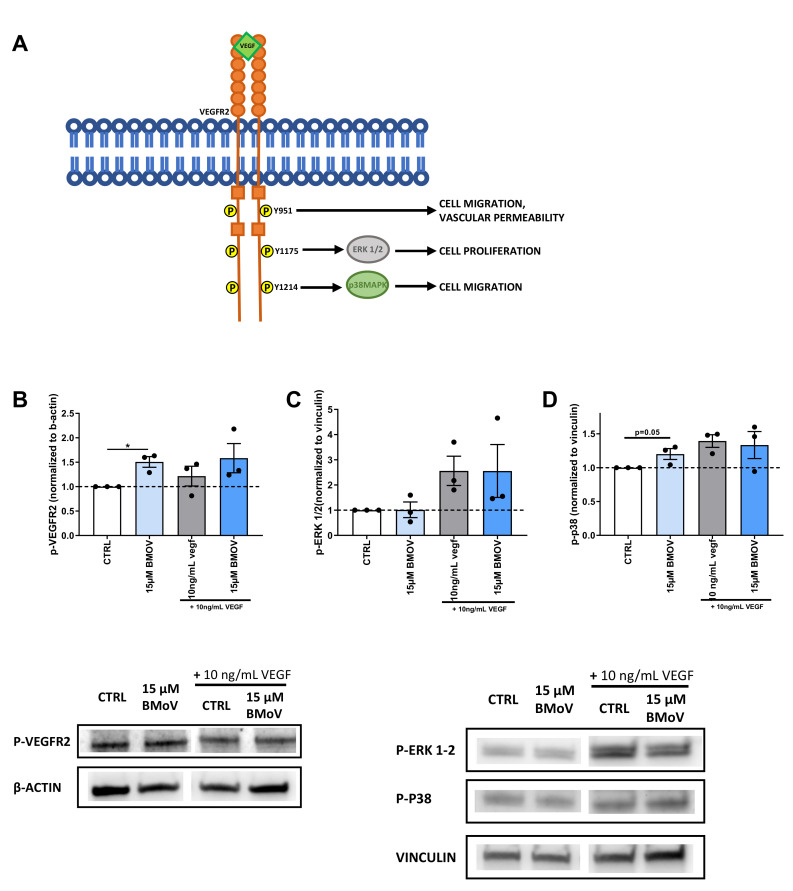
(**A**) Schematic representation of VEGFR2 signaling and downstream enzyme’s activation. (**B**) Relative p-VEGFR2, protein expression in HUVECs whole-cell lysates after treatment with indicated conditions as determined by western blot. Expression was normalized per independent experiment to stable household protein β-actin and expressed relative to CTRL. Relative (**C)** p-ERK1/2 and (**D**) p-p38, protein expression in HUVECs whole-cell lysates after treatment with indicated conditions as determined by western blot. Expression was normalized per independent experiment to stable household protein vinculin and expressed relative to CTRL. (E) Quantification of the concentration of VEGF-A concentration in the cell culture medium of HUVECs incubated with the indicated conditions. All data points represent normalized averages obtained from 3 independent experiments and are presented as mean ± SEM. **p* < 0.05; by one-sample t test versus CTRL or two-sided Student’s t test to compare CTRL versus BMOV treatments.

**Figure 9 ijms-21-04643-f009:**
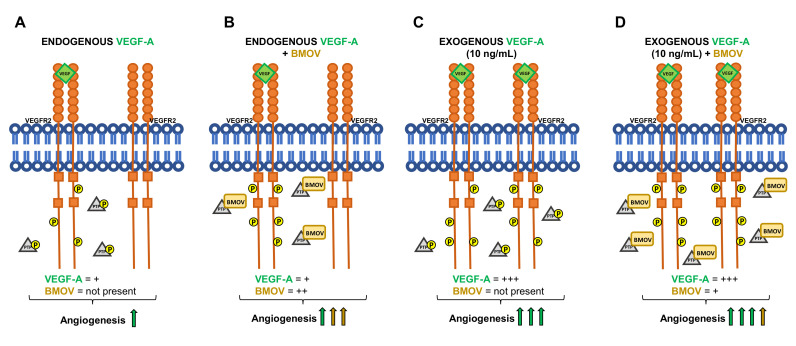
Schematic representation of the results obtained. (**A**) VEGF-A endogenously produced by HUVECs activates VEGFR2 resulting in its partial phosphorylation and low levels of angiogenesis. No addition of BMOV is carried out in this condition. (**B**) VEGFR2 is activated by endogenous VEGF-A and BMOV, by blocking PTPs, induces an increase in VEGFR2 phosphorylation that results in increased angiogenesis. (**C**) Exogenous addition of VEGF-A (10 ng/mL) results in increased angiogenesis via increased VEGFR2 phosphorylation. No addition of BMOV is carried out in this condition. (**D**) Coadministration of exogenous VEGF-A (10 ng/mL) and BMOV results in increased angiogenesis. The effect of BMOV is indicated with yellow arrow, the effect of VEGF-A is indicated with green arrows.

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
