# Peer review of "Bis(maltolato)oxovanadium(IV) Induces Angiogenesis via Phosphorylation of VEGFR2"

_ijms, 2020, doi:10.3390/ijms21134643_

Round 1

Reviewer 1 Report

This is a well designed and conducted study and a beautifully written manuscript identifying BMOV as a pro-angiogenic agent.

I only have one minor suggestion:it would be useful to include information on how the BMOV doses were selected, particularly for the in vivo studies.

Reviewer 2 Report

The manuscript submitted by Parma and coworkers study the role of BMOV as an activator or stimulator of angiogenesis, which may ultimately result of interest for the potential treatment of peripheral arterial disease (PAD). They found that BMOV induces wound closure in vivo by 45%, and further studied its effect on cell migration, proliferation and tube formation in vitro. The authors also observed a 3-fold increase in cell sprouting using an ex vivo aortic ring assay. Furthermore, BMOV did not show a synergistic effect in combination with exogenous VEGF-A, and phosphorylation of Y951 and Y1214 residues pointed towards the participation of VEGFR-2 and p38 MAPK enzyme into the pro-angiogenesis mechanism of action of BMOV.

The work is interesting and of sufficient novelty and importance to justify its publication in Int. J. Mol. Sci. Some specific comments are given below.

  • BMOV has been previously studied as an inhibitor of PTP for the treatment of diabetes and cancer. However and although the ability of BMOV to bind and inhibit the PTP enzymes has led the author to study its angiogenic effects, this is not well described in the introduction. In addition, the importance of angiogenesis stimulators in the biomedical context is not described until the Discussion section, which makes difficult to understand the applicability of compounds with such activity. Please add a sentence or two to put it into context.
  • The author’s hypothesis is outlined in Figure 2A. I recommend making it an independent Figure 1. The chemical structure of BMOV must be added in the Figure.
  • The time at which the experiments were done is not specified within the Results section. Please clarify in the text at what time the VEGF-A quantification, cell migration, MTT and tube formation were done.
  • The authors used two different graphs (normalized to control and normalized to 10 ng/ml hVEGF) to show the cumulative effect of BMOV when combined with VEGF-A. However, both graphs represent the same data so the inclusion of the second is redundant. One graph (normalized to control) that includes the control, 10 ng/ml hVEGF as well as the different doses of BMOV would be clearer to the reader. Please, just place the dotted line on top of the 10 ng/ml hVEGF column to show the added effect of VEGF-A.
  • Although the abbreviation BMOV is used through the manuscript, BMoV is used on the images as well as in the Figures. Please choose one and make it consistent.
  • Revise Paragraph 122-126. Figure 3A shows a 45% increase of cell migration, however in the next paragraph it is said that BMOV gave a 37% increase, leading the reader to the same Figure (Figure 3A). I guess, there is a mistake and the 37% increase is coming from the exogenous VEGF-A, which in combination to the 41% of BMOV gives the total of 78% found when BMOV is combined with VEGF-A.
  • Line 294: What do the authors mean with “BMOV can not add to this effect”? What effect? Please make the sentence clearer.
  • Clarify in “BMOV preparation” in the Material and methods section whether BMOV was prepared by the authors, bought or given by someone.
  • I suggest the authors to carefully revise the writing of the paper, and to avoid the excessive use of expressions like “we obtained”, “we here found”, “we here demonstrate”…Some detected mistakes are:
    1. Line 68: “…that show increased availability”. Change availability for bioavailability or availability in vivo.
    2. Line 135: Change “…and what would to this effect” for “…and what would do to this effect”.
    3. Line 255: Change “Combined these results…” for “ Combined, these results…”
    4. Line 300: Change “depending” for “dependent”
    5. Line 321: Change “…by shaving and hair removal cream.” for 
“…by shaving using hair removal cream.”
    6. Line 324: Change “iv” for “intravenous (iv)”
    7. Line 401: “96. -well plates” for “96 well plates”
  • The references are written using either the full name of the journal or their abbreviation indistinctively. Please, make the References section consistent.
